# COUPLED ENSEMBLES OF NEURAL NETWORKS

## ABSTRACT

We investigate in this paper the architecture of deep convolutional networks. Building on existing state of the art models, we propose a reconfiguration of the model parameters into several parallel branches at the global network level, with each branch being a standalone CNN. We show that this arrangement is an efficient way to significantly reduce the number of parameters while at the same time improving the performance. The use of branches brings an additional form of regularization. In addition to splitting the parameters into parallel branches, we propose a tighter coupling of these branches by averaging their log-probabilities. The tighter coupling favours the learning of better representations, even at the level of the individual branches, as compared to when each branch is trained independently. We refer to this branched architecture as "coupled ensembles". The approach is generic and can be applied to almost any neural network architecture. With coupled ensembles of DenseNet-BC and parameter budget of 25M, we obtain error rates of 2.92%, 15.68% and 1.50% on CIFAR-10, CIFAR-100 and SVHN respectively. For the same parameter budget, DenseNet-BC has an error rate of 3.46%, 17.18%, and 1.8% respectively. With ensembles of coupled ensembles, of DenseNet-BC networks, with 50M total parameters, we obtain error rates of 2.72%, 15.13% and 1.42% respectively on these tasks.

## 1 INTRODUCTION

The design of early convolutional architectures (CNN) involved choices of hyper-parameters such as: filter size, number of filters at each layer, and padding (LeCun et al., 1998; Krizhevsky et al., 2012). Since the introduction of the VGGNet (Simonyan & Zisserman, 2014) the design has moved towards following a template: fixed filter size of $3 \times 3$ and $N$ features maps, down-sample to half the input resolution *only* by the use of either `maxpool` or strided convolutions (Springenberg et al., 2015), and double the number the computed feature maps following each down-sampling operation. This philosophy is used by state of the art models like ResNet (He et al., 2016b) and DenseNet (Huang et al., 2017b). The last two architectures extended the template to include the use of "skip-connections" between non-contiguous layers.

Our work extends this template by adding another element, which we refer to as "coupled ensembling". In this set-up, the network is decomposed into several branches, each branch being functionally similar to a complete CNN. The proposed template achieves performance comparable to state of the art models with a significantly *lower* parameter count.

In this paper, we make the following contributions: (i) we show that given a parameter budget, it is better to have the parameters split among branches rather than having a single branch (which is the case for all current networks), (ii) we compare different ways to combine the activations of the parallel branches and find that it is best to take an arithmetic mean of the individual log-probabilities (iii) combining these elements, we significantly match and improve the performance of convolutional networks on CIFAR and SVHN datasets, with a heavily reduced parameter count. (iv) Further ensembling of coupled ensembles lead to more improvement. This paper is organised as follows: in section 2, we discuss related work; in section 3, we introduce the concept of coupled ensembles and the motivation behind the idea; in section 4, we evaluate the proposed approach and compare it with the state of the art; and we conclude and discuss future work in section 5.

## 2    RELATED WORK

**Multi-column architectures.** The network architecture that we propose has strong similarities with Cireşan's Neural Networks Committees (Cireşan et al., 2011) and Multi-Column Deep Neural Network (MCDNN) (Cireşan et al., 2012), which are a type of ensemble of networks where the "committee members" or "DNN columns" correspond to our element blocks (or branches). However, our coupled ensemble networks differ as following: (i) we train a *single* model which is *composed* of branches, while they train each member or column *separately*. (ii) we have a fixed parameter budget for the *entire* model for improving the performance. This is contrary to improving it by utilising multiple models of fixed size and therefore multiplying the overall size (though both are not exclusive); (iii) we combine the activations of the branches by combining their log-probabilities over the target categories, and (iv) we used the same input for all branches while they considered different preprocessing (data augmentation) blocks for different members or different subsets of columns.

**Multi-branch architectures.** Multi-branch architectures have been very successful in several vision applications (He et al., 2016a; Szegedy et al., 2015). Recently, modifications have been proposed (Xie et al., 2017; Chollet, 2017) for these architectures using the concept of "grouped convolutions", in order to factorize spatial and depth wise feature extraction. These modifications additionally advocate the use of *template* building blocks stacked together to form the complete model. This modification is at the level of the *building blocks* of their corresponding *base* architectures: ResNet and Inception respectively. In contrast we propose a generic modification of the structure of the CNN at the global model level. This includes a template in which the specific architecture of a "element block" is specified, and then this "element block" is replicated as parallel branches to form the final composite model.

To further improve the performance of such architectures, Shake-Shake regularization (Gastaldi, 2017) proposes a stochastic mixture of each of the branches and has achieved good results on the CIFAR datasets. However, the number of epochs required for convergence is much higher compared to the base model. Additionally, the technique seems to depend on the batch size. In contrast, we apply our method using the exact *same* hyper-parameters as used in the underlying CNN.

Zhao et al. (2016) investigate the usage of parallel paths in a ResNet, connecting layers across paths to allow information exchange between them. However this requires modification at a local level of each of the residual blocks. In contrast, our method is a generic rearrangement of a given architecture's parameters, which does not introduce additional choices. Additionally, we empirically confirm that our proposed configuration leads to an efficient usage of parameters.

**Neural network ensembles.** Ensembling is a reliable technique to increase the performance of models for a task. Due to the presence of several local minima, multiple trainings of the exact same neural network architecture can reach a different distribution of errors on a per-class basis. Hence, combining their outputs lead to improved performance on the overall task. This was observed very early (Hansen & Salamon, 1990) and is now commonly used for obtaining top results in classification challenges, despite the increase in training and prediction cost. Our proposed model architecture is not an ensemble of independent networks given that we have a single model made up of parallel branches that is trained jointly. This is similar in spirit to the `residual block` in ResNet and ResNeXt, and to the `inception` module in Inception but is done at the global network level. We would like to emphasize here that "arranging" a given budget of parameters into parallel branches leads to an increase in performance (Tables 1, 2, 3). Additionally, the classical ensembling approach can still be applied for the fusion of independently trained coupled ensemble models, where it leads to a significant performance improvement (Table 4).

**Snapshot ensembles.** Huang et al. (2017a) and Loshchilov & Hutter (2017) used the ensembling approach on checkpoints during the training process instead of using fully converged models. This approach is quite efficient since the obtained performance is higher for a same training time budget. However, both the overall model size and prediction time are significantly increased. Given a model size and performance measure, our approach aims to either keep the model size constant and improves the performance, or to obtain the same performance with a smaller model size.

## 3   COUPLED ENSEMBLES

TERMINOLOGY

For the following discussion, we define some terms:

- Branch: the proposed model comprises several branches. The number of branches is denoted by $e$ Each branch takes as input a data point and produces a score vector corresponding to the target classes. Current design of CNNs are referred to as single-branch ($e = 1$).

- Element block: the model architecture used to form a branch. In our experiments, we use DenseNet-BC and ResNet with pre-activation as element blocks.

- Fuse Layer: the operation used to combine each of the parallel branches which make up our model. In our experiments, each of the branches are combined by taking the average of each of their individual log probabilities over the target classes. Section 4.4 explores different choices of operations for the fuse layer.

We consider a classification task in which individual samples are always associated to exactly one class, labelled from a finite set. This is the case for CIFAR (Krizhevsky & Hinton, 2009), SVHN (Netzer et al., 2011) and ILSVRC (Russakovsky et al., 2015) tasks. In theory, this generalises to other tasks as well (for example, segmentation, object detection, etc.).

We consider neural network models which output a score vector of the same dimension as the number of target classes. This is usually implemented as a linear layer and referred to as a fully connected (FC) layer. This layer can be followed by a SoftMax (SM) layer to produce a probability distribution over the target classes. During training, this is followed by a loss layer, for example, negative log-likelihood (LL). This is the case for all recent network architectures for image classification[1] (Krizhevsky et al., 2012; Simonyan & Zisserman, 2014; Szegedy et al., 2015; He et al., 2016a; Xie et al., 2017; Huang et al., 2016; Zagoruyko & Komodakis, 2016; Huang et al., 2017b). The differences among them is related to what is present before the last FC layer. We are agnostic to this internal setup (however complex it may or may not be) because the resulting "element block" always takes an image as input and produces a vector of $N$ values as output, parametrized by a tensor $W$.

In the case of ensemble of independently trained models, fusion is usually done by computing the individual predictions separately for $e$ model instances and then averaging them. Each of the instances are trained *seperately*. This is functionally equivalent to predicting with a "super-network" including the $e$ instances as parallel branches with an averaging (AVG) layer on top. Such supernetworks are generally not implemented because the branch instances often already correspond to the maximum memory capacity of GPUs. The remaining AVG layer operation can be implemented separately. Alternatively, it is possible to place the averaging layer just after the last FC layer of the element block instances and before the SM layer, which is then "factorized".

In our set-up, the model is composed of parallel branches and each branch produces a score vector for the target categories. The score vectors are fused through the "fuse layer" during training and the composite model produces a single prediction. We refer to this as coupled ensembles (fig. 1).

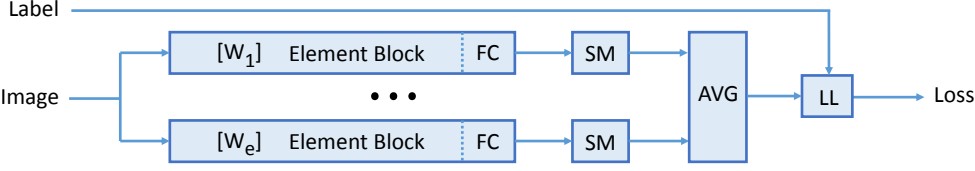

Figure 1: Train versions of coupled ensemble networks (here with averaging at the level of the (Log)SoftMax layer).

---

[1]Though these networks actually do have a FC layer before the SM one, the last layer need not be a linear layer, as long as it produces one value for each target label. We will refer to the output of each element block as "FC". Similarly, the proposed method may be easily adapted to multi-label classification by simply replacing the SM (and possibly also the LL) layer(s) by any variant(s) appropriate for multi-label classification. Again, we will refer to their output as "SM" and "LL".

We explore three options to combine score vectors during training and for inference (section 4.4):

- Activation (FC) average: Average the output of the FC layers of the branches
- Probability (LSM) average: Average the log-probabilities that each branch predicts for the target classes.
- Log Likelihood (LL) average: Average the loss of each branch.

Note that for inference, averaging the FC layer activations is equivalent to averaging the log-probabilities (see section C of supplementary material).

This transformation of having multiple branches, and combining the branch score vectors by averaging the log probabilities of the target categories, leads to a performance improvement, with a *lower* parameter count, in all our experiments (see section 4). The parameter vector $W$ of this composite branched model is the concatenation of the parameter vectors $W_e$ of the $e$ element blocks with $1 \leq i \leq e$. All parameters are in the "element blocks" as the "fuse layer" does not contain any parameters.

# 4 EXPERIMENTS

## 4.1 DATASETS

We evaluate our proposed architecture on the CIFAR (Krizhevsky & Hinton, 2009) and SVHN (Netzer et al., 2011) data sets. CIFAR-10 and CIFAR-100 consist of 50,000 training images and 10,000 test images, categorised into 10 and 100 categories respectively. SVHN consists of 73,257 training images, 531,131 "easy" training images (we use both for training) and 26,032 testing images, categorised into 10 categories. All images have a size of 32×32 pixels.

## 4.2 EXPERIMENTAL SETUP

All hyper parameters are set according to the original descriptions of the "element block" that is used. This may not be the optimal setting in our case (especially the learning rate decay schedule) but we do not alter them, so as to not introduce any bias in comparisons.

For CIFAR-10, CIFAR-100 and SVHN, the input image is normalised by subtracting by the mean image and dividing by the standard deviation. During training on CIFAR datasets, standard data augmentation is used, which comprises random horizontal flips and random crops. For SVHN, no data augmentation is used. However, a dropout ratio of 0.2 is applied in the case of DenseNet when training on SVHN. Testing is done after normalising the input in the same way as during training.

All error rates are given in percentages and correspond to an average of the last 10 epochs. This measure is more conservative than the one used by the DenseNet authors (see supplementary material, section F). For DenseNet-BC, Amos & Kolter (2017)'s PyTorch implementation has been used. All execution times were measured using a single NVIDIA 1080Ti GPU with the optimal micro-batch[2]. Experiments in section 4.3, 4.4 are done on the CIFAR-100 data set with the "element block" being DenseNet-BC, depth $L = 100$, growth rate $k = 12$. For experiments in Section 4.5, we consider this same configuration (with a single branch) as our baseline reference point.

## 4.3 COMPARISON WITH SINGLE BRANCH AND INDEPENDENT ENSEMBLES

A natural point of comparison of the proposed branched architecture is with an ensemble of independent models. Rows 2 (coupled ensemble with LSM averaging) and 4 (independent training) in Table 1 present the results of these two cases respectively. Row 4 shows the error rate obtained from averaging the predictions of 4 identical models, each trained separately. We see that even though the total number of trainable parameters involved is exactly the same, a *jointly trained branched* configuration gives a much lower test error (17.61 vs. 18.42).

The next point of comparison is with a *single* branch model comprising a similar number of parameters as the multi branch configuration. The choice of single branch models has been done by:

---

[2]See section B

increasing $k$ while keeping $L$ constant, by increasing both $k$ and $L$, or by increasing $L$ while keeping $k$ constant. The last three rows of Table 1 show that the error from the multi branch model is considerably lower, as compared to a single branch model (17.61 vs. 20.01). This shows that the effect of the branched configuration is more pronounced as the number of parameters increase (Section D.1 has additional results).

These observations show that the arranging a given budget of parameters into parallel branches is more efficient in terms of parameters, as compared to having a large single branch or multiple independent models. In Section 4.5 we analyse the relation between the number of branches and the model performance.

## 4.4 CHOICE OF FUSE LAYER OPERATION

In this section, we compare the performance of our proposed branched model for different choices of the "fuse layer" (see section 3). Experiments are carried out to evaluate the best training and prediction fusion combinations. We consider a branched model with $e = 4$, trained in the following conditions: training with fusion after the FC layer, after the LSM layer, or after the LL layer.

Table 1: Coupled Ensembles of DenseNet-BCs ($e = 4$) with different "fuse layer" combinations versus a single branch model. Performance is given as the top-1 error rate (mean±standard deviation for the individual branches) on the CIFAR-100 test set. Columns "$L$" and "$k$" denote the "element block" architecture, "$e$" is the number of branches. Column "Avg." indicates the type of "fuse layer" during training (section 3); "none" for separate trainings. Column "Individual" is error of each branch; Columns "FC" and "SM" give the performance for "fuse layer" choices during inference. Epoch is time taken to complete one training epoch, Test is time for testing each image. (*) See supplementary material, section F; The average and standard deviations are computed here for the independent trainings (comprising 4 models).

| $L$ | $k$ | $e$ | Avg. | Individual | FC | SM | Params | Epoch(s) | Test(ms) |
|---|---|---|---|---|---|---|---|---|---|
| 100 | 12 | 4 | FC | 74.36±26.28 | 22.55 | 31.92 | 3.20M | 402 | 2.00 |
| 100 | 12 | 4 | LSM | 22.29±0.11 | **17.61** | 17.68 | 3.20M | 402 | 2.00 |
| 100 | 12 | 4 | LL | 22.83±0.18 | 18.21 | 18.92 | 3.20M | 402 | 2.00 |
| 100 | 12 | 4 | none | 23.13±0.09(*) | 18.42 | 18.85 | 3.20M | 341 | 2.00 |
| 100 | 25 | 1 | n/a | 20.61±0.01 | n/a | n/a | 3.34M | 164 | 0.8 |
| 154 | 17 | 1 | n/a | 20.02±0.10 | n/a | n/a | 3.29M | 245 | 1.3 |
| 220 | 12 | 1 | n/a | 20.01±0.12 | n/a | n/a | 3.15M | 326 | 1.5 |

Table 1 shows the performance of models under different "fuse layer" operations for inference. *Note that this table includes models with parameters obtained using different training methods* . We can make the following observations:

- The branched model with $e = 4$ and Avg. LSM for the "fuse layer" has almost the *same* performance as a DenseNet-BC ($L = 250, k = 24$) model (Huang et al., 2017b), which has about *5 times more* parameters (15.3M versus 3.2M).

- The average error rate of each of the "element blocks" trained jointly in coupled ensembles with LSM fusion is significantly lower than the error rate of the individual instances trained separately. This indicates that the coupling not only forces them to learn complementary features as a group but also to learn better representations, individually. Averaging the log probabilities forces the network to continuously update all branches so as to be consistent with each other. This provides a stronger gradient signal. The error gradient that is back-propagated from the fuse layer is the same for all branches, and this gradient depends on the *combined* predictions. This means that at every step all branches act complementary to the other branches' weight updates.

- All ensemble combinations except the Avg. FC training do significantly better than a single branch network. For a parameter budget of about 3.2M, the error rate of the best single branch model is 20.01. Keeping the same parameter budget, using 4 branches reduces the error rate to 17.61 ($-2.40$).

- When training with Avg. FC, the individual branches do not perform well (in red). This is expected since a similar FC average may be reached with quite unrelated FC instances. The Avg. FC training with Avg. SM prediction (in red) works a bit better but is still not good because the non-linearity of the SM layer distorts the FC average. Avg. FC training with Avg. FC prediction works quite well though it does not yield the best performance.

- The Avg. FC prediction works at least as well and often significantly better than the Avg. SM prediction. This can be explained by the fact that the SM layer normalises values to probabilities (between 0 and 1), while the FC values remain spread and transmit more information at the FC layer.

All the following experiments have Avg. LSM for the training "fuse layer" in the branched models.

### 4.5 CHOICE OF THE NUMBER OF BRANCHES

In this section, we investigate the optimal number of branches $e$ for a given model parameter budget. We evaluate on CIFAR-100, with DenseNet-BC as the "element block", and parameter budget equal to $0.8M$ (number of parameters in DenseNet-BC ($L = 100, k = 12$)). Indeed, the optimal number of instances $e$ is likely to depend upon the network architecture, upon the parameter budget and upon the data set but this gives at least one reference. This was investigated again with larger models, and the results are in table 3 (last four rows).

Table 2: Different number of branches, $e$, for a parameter budget. The models are trained on CIFAR-100 with standard data augmentation. See table 1 caption for the meaning of row and column labels. When applicable ($e > 1$), "fuse layer" is LSM Avg. (*) Average and standard deviation on 10 trials with different seeds; Huang et al. (2017b) reports 22.27, see supplementary material, section F.

| $L$ | $k$ | $e$ | Individual | FC | SM | Params | Epoch(s) | Test(ms) |
|-----|-----|-----|------------|-----|-----|--------|----------|----------|
| 100 | 12 | 1 | **22.87±0.17(\*)** | n/a | n/a | 800k | 86 | 0.51 |
| 76 | 10 | 2 | 25.58±0.20 | 21.66 | 22.17 | 720k | 103 | 0.63 |
| 82 | 8 | 3 | 26.47±0.17 | **21.25** | 21.46 | 800k | 141 | 0.85 |
| 70 | 8 | 4 | 27.65±0.48 | 21.50 | 22.12 | 828k | 156 | 0.94 |
| 64 | 7 | 6 | 30.65±0.62 | 23.08 | 23.36 | 840k | 198 | 1.20 |
| 64 | 6 | 8 | 31.52±0.38 | 24.42 | 24.69 | 843k | 250 | 1.51 |

Table 2 shows the performance for different configurations of branches $e$, depth $L$, and growth rate $k$. One difficulty is that DenseNet-BC parameter counts are strongly quantified according to the $L$ and $k$ values ($L$ has to be a multiple of 6 modulo 4) and, additionally, to the $e$ value in the coupled ensemble version. This is even more critical in moderate size models like the 800K one targeted here. We selected model configurations with parameters just below the target to have a fair comparison. A few models have slightly more parameters so that some interpolation can be done for possibly more accurate comparisons. We can make the following observations:

- In the considered case (DenseNet-BC, CIFAR-100 and 800k parameters), the optimal number of branches is $e = 3, L = 70, k = 9$. With this configuration, the error rates decreases from 22.87 for the single branch ($L = 100, k = 12$) DenseNet-BC model to 21.10 ($-1.77$).

- Using 2 to 4 branches yields a significant performance gain over the classical (single branch, $e = 1$) case, and even over the original performance of 22.27 reported for the ($L = 100, k = 12$) DenseNet-BC (see supplementary material, section F).

- Using 6 or 8 branches performs significantly worse, possibly because the element blocks are too 'thin'.

- Model performance is robust to slight variations of $L$, $k$ and $e$ around their optimal values, showing that the coupled ensemble approach and the DenseNet-BC architecture are quite robust relative to these choices.

- The gain in performance comes at the expense of an increased training and prediction times even though the model size does not change. This is due to the use of smaller values of $k$ that reduces the throughput for smaller models.

### 4.6 Comparison with state of the art

We have evaluated coupled ensembles against existing models of various sizes. We used again Huang et al. (2017b)'s DenseNet-BC architecture as the "element block" since this was the current state of the art or very close to it at the time we started these experiments. We also evaluated ResNet He et al. (2016b)'s with pre-activation as the element block to check if the coupled ensemble approach works well with other architectures.

Table 3 shows the current state of the art models (see section 2 for references) and performance of coupled ensembles in the lower part. All results presented in this table correspond to the predictions of single model. If ensembles are considered, they are always coupled as described in section 3 and trained as a single global model. A further level of ensembling involving multiple models is considered in section 4.7.

Table 3: Classification error comparison with the state of the art, for single model training.

| Architecture | C10+ | C100+ | SVHN | #Params |
|---|---|---|---|---|
| ResNet $L = 110$ $k = 64$ (He et al., 2016a) | 6.61 | - | - | 1.7M |
| ResNet stochastic depth $L = 110$ $k = 64$ | 5.25 | 24.98 | - | 1.7M |
| ResNet stochastic depth $L = 1202$ $k = 64$ | 4.91 | - | - | 10.2M |
| ResNet pre-act. $L = 164$ $k = 64$ (He et al., 2016b) | 5.46 | 24.33 | - | 1.7M |
| ResNet pre-act. $L = 1001$ $k = 64$ | 4.92 | 22.71 | - | 10.2M |
| DenseNet $L = 100$ $k = 24$ (Huang et al., 2017b) | 3.74 | 19.25 | 1.59 | 27.2M |
| DenseNet-BC $L = 100$ $k = 12$ (Huang et al., 2017b) | 4.51 | 22.27 | 1.76 | 0.80M |
| DenseNet-BC $L = 250$ $k = 24$ | 3.62 | 17.60 | - | 15.3M |
| DenseNet-BC $L = 190$ $k = 40$ | 3.46 | 17.18 | - | 25.6M |
| Shake-Shake C10 Model S-S-I (Gastaldi, 2017) | 2.86 | - | - | 26.2M |
| Shake-Shake C100 Model S-E-I | - | 15.85 | - | 34.4M |
| Snapshot Ensemble DenseNet-40 ($\alpha_0 = 0.1$) | 4.99 | 23.34 | 1.64 | 6.0M |
| Snapshot Ensemble DenseNet-40 ($\alpha_0 = 0.2$) | 4.84 | 21.93 | 1.73 | 6.0M |
| Snapshot Ensemble DenseNet-100 ($\alpha_0 = 0.2$) | 3.44 | 17.41 | - | 163M |
| SGDR WRN-28-10 (Loshchilov & Hutter, 2017) | 4.03 | 19.57 | - | 36.5M |
| SGDR WRN-28-10 3 snapshots | 3.51 | 17.75 | - | 110M |
| ResNeXt-29, 8×64d (Xie et al., 2017) | 3.65 | 17.77 | - | 34.4M |
| ResNeXt-29, 16×64d | 3.58 | 17.31 | - | 68.1M |
| DFN-MR2 (Zhao et al., 2016) | 3.94 | 19.25 | 1.51 | 14.9M |
| DFN-MR3 | 3.57 | 19.00 | 1.55 | 24.8M |
| IGC-L450M2 (Zhang et al., 2017) | 3.25 | 19.25 | - | 19.3M |
| IGC-L32M26 | 3.31 | 18.75 | 1.56 | 24.1M |
| ResNet pre-activation $L = 65$ $k = 64$ $e = 2$ | 5.26 | 23.24 | - | 1.4M |
| ResNet pre-activation $L = 164$ $k = 64$ $e = 2$ | 4.24 | 19.92 | - | 3.4M |
| ResNet pre-activation $L = 164$ $k = 64$ $e = 4$ | 3.96 | 18.84 | - | 6.8M |
| DenseNet-BC $L = 100$ $k = 12$ $e = 1$ | 4.77 | 22.87 | 1.79 | 0.8M |
| DenseNet-BC $L = 112$ $k = 16$ $e = 1$ | 4.47 | 20.73 | 1.83 | 1.7M |
| DenseNet-BC $L = 130$ $k = 20$ $e = 1$ | 3.86 | 19.62 | 1.84 | 3.4M |
| DenseNet-BC $L = 160$ $k = 24$ $e = 1$ | 3.74 | 18.43 | 1.88 | 6.9M |
| DenseNet-BC $L = 166$ $k = 32$ $e = 1$ | 3.68 | 17.68 | 1.88 | 13.0M |
| DenseNet-BC $L = 190$ $k = 40$ $e = 1$ | 3.75 | 17.22 | 1.79 | 25.8M |
| DenseNet-BC $L = 82$ $k = 8$ $e = 3$ | 4.30 | 21.25 | 1.66 | 0.8M |
| DenseNet-BC $L = 82$ $k = 10$ $e = 4$ | 3.78 | 19.92 | 1.62 | 1.6M |
| DenseNet-BC $L = 88$ $k = 14$ $e = 4$ | 3.57 | 17.68 | 1.55 | 3.5M |
| DenseNet-BC $L = 88$ $k = 20$ $e = 4$ | 3.18 | 16.79 | 1.57 | 7.0M |
| DenseNet-BC $L = 94$ $k = 26$ $e = 4$ | 3.01 | 16.24 | 1.50 | 13.0M |
| DenseNet-BC $L = 118$ $k = 35$ $e = 3$ | 2.99 | 16.18 | 1.50 | 25.7M |
| DenseNet-BC $L = 106$ $k = 33$ $e = 4$ | 2.99 | 15.68 | 1.53 | 25.1M |
| DenseNet-BC $L = 76$ $k = 35$ $e = 6$ | 2.92 | 15.76 | 1.50 | 24.6M |
| DenseNet-BC $L = 64$ $k = 35$ $e = 8$ | 3.13 | 15.95 | 1.50 | 24.9M |

Coupled ensembles with ResNet `pre-act` as element block and $e = 2, 4$ leads to a significantly better performance than single branch models, which have comparable or *higher* number of parameters.

For the DenseNet-BC architecture, we considered 6 different network sizes, ranging from 0.8M up to 25.6M parameters. Huang et al. (2017b) reports results for the two extreme cases. We chose these values for the depth $L$ and growth rate $k$ for these points and interpolated between them according to a log scale as much as possible. Our experiments show that the trade-off between $L$ and $k$ is not critical for a given parameter budget. This was also the case for choosing between the number of branches $e$, depth $L$ and growth rate, $k$ for a fixed parameter budget as long as $e \geq 3$ (or even $e \geq 2$ for small networks). For the 6 configurations, we experimented with both the single-branch (classical) and multi-branch versions of the model, with $e = 4$. Additionally, for the largest model, we tried $e = 3, 6, 8$ branches.

For single branch DenseNet-BC, we obtained error rates higher than reported by Huang et al. (2017b). From what we have checked, their Torch7 implementation and our PyTorch one are equivalent. The difference may be due to the fact that we used a more conservative measure of the error rate (on the last iterations) and from statistical differences due to different initializations and/or to non-deterministic computations (see section F in supplementary material). Still, the coupled ensemble leads to a significantly better performance for all network sizes, even when compared to DenseNet-BC's reported performance.

Our larger models of coupled DenseNet-BCs (error rates of 2.92% on CIFAR 10, 15.68% on CIFAR 100 and 1.50% on SVHN) perform better than or are on par with all current state of the art implementations that we are aware of at the time of submission of this work. Only the Shake-Shake S-S-I model (Gastaldi, 2017) performs slightly better on CIFAR 10.

We also compare the performance of coupled ensembles with model architectures that were 'learnt' in a meta learning scenario. The results are presented in the supplementary material, section E.

## 4.7 ENSEMBLES OF COUPLED ENSEMBLES

The coupled ensemble approach is limited by the size of the network that can fit into GPU memory and the training time. With the hardware we have access to, it was not possible to go much beyond the 25M parameters. For going further, we resorted to the classical ensembling approach based on independent trainings. An interesting question was whether we could still significantly improve the performance since the classical approach generally plateaus after quite a small number of models and the coupled ensemble approach already include several. For instance, SGDR with snapshots (Loshchilov & Hutter, 2017) has a significant improvement from 1 to 3 models but not much improvement from 3 to 16 models (see tables 3 and 4). As doing multiple times the same training is quite costly for large, we instead ensembled the four large coupled ensemble models, $e = 3, 4, 6, 8$. Results are shown in table 4. We obtained a significant gain by fusing two models and a quite small one from any further fusion of three or four of them. To the best of our knowledge, these ensembles of coupled ensemble networks outperform all state of the art implementations including other ensemble-based ones at the time of submission of this work.

Table 4: Classification error comparison with the state of the art, multiple model trainings.

| Architecture | C10+ | C100+ | SVHN | #Params |
|---|---|---|---|---|
| SGDR WRN-28-10 3 runs $\times$ 3 snapshots | 3.25 | 16.64 | - | 329M |
| SGDR WRN-28-10 16 runs $\times$ 3 snapshots | 3.14 | 16.21 | - | 1752M |
| DenseNet-BC ensemble of ensembles $e = 6, 4$ | 2.72 | 15.13 | 1.42 | 50M |
| DenseNet-BC ensemble of ensembles $e = 6, 4, 3$ | 2.68 | 15.04 | 1.42 | 75M |
| DenseNet-BC ensemble of ensembles $e = 8, 6, 4, 3$ | 2.73 | 15.05 | 1.41 | 100M |

## 4.8 PARAMETER USAGE

The lower half of table 3, the sections with $e = 1$ and $e > 1$ highlight the difference in error rates between single and multi branch models. A branched model with 13M parameters has an error rate of 16.24 for CIFAR-100. In contrast, none of the single branch models match this performance even with double the number of parameters. Figure 2 compares this with other state of the art models.

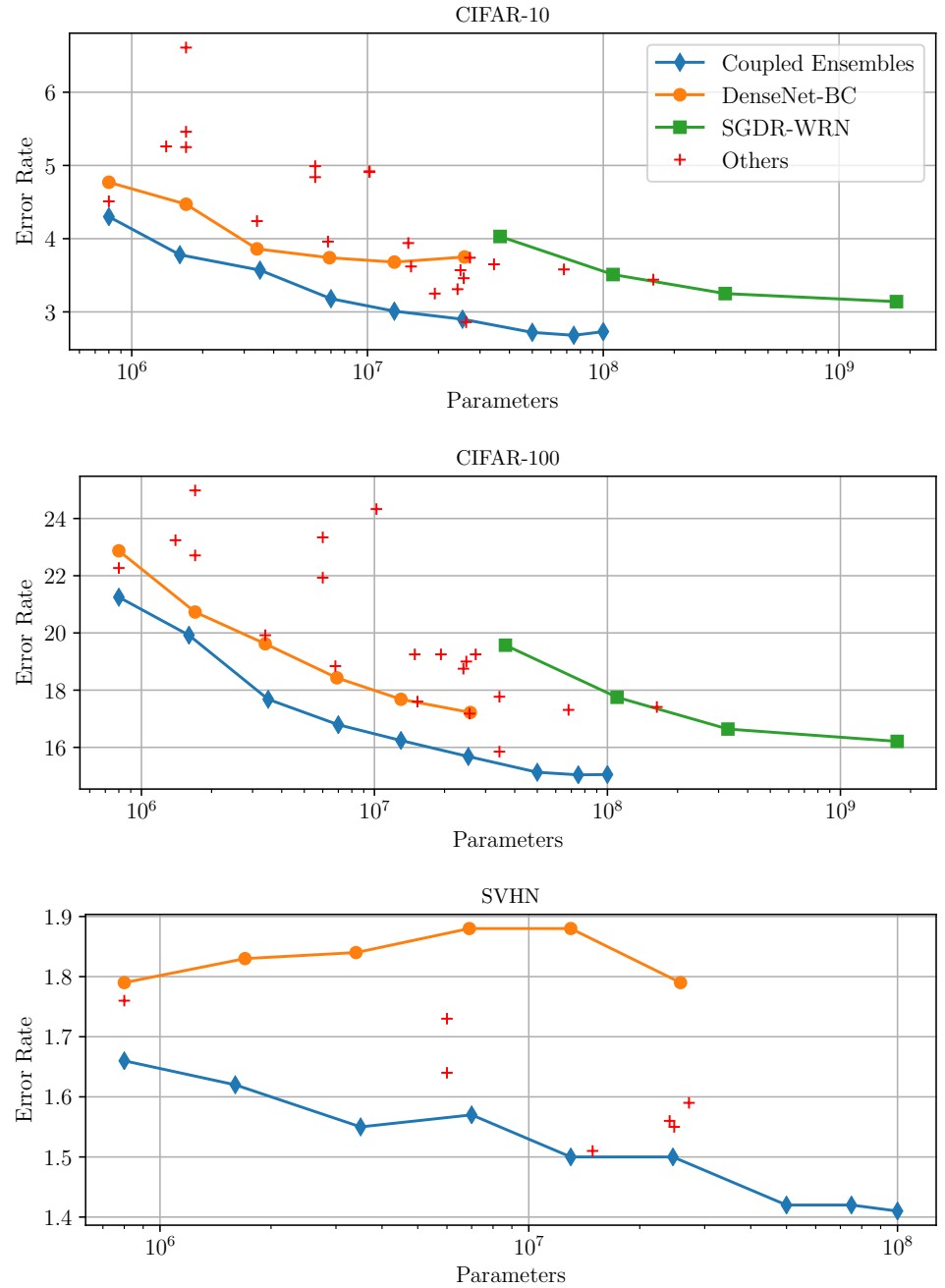

Figure 2: Comparison of parameter usage and error rate among different methods, for CIFAR-10, 100 and SVHN. "Ours": coupled ensemble with DenseNet-BC "element block". Single model up to 25M parameters and ensembles of coupled ensembles beyond; SGDR-WRN: snapshot ensembles up to 110M parameters and ensembles of snapshot ensembles beyond; "Other": all other architectures from tables 3, 4.

## 5 DISCUSSION

The proposed approach consists in replacing a single deep convolutional network by a number of "element blocks" which resemble standalone CNN models. The intermediate score vectors produced by each of the elements blocks are coupled via a "fuse layer". At training time, this is done by taking an arithmetic average of their log-probabilities for the targets. At test time the score vectors are averaged following the output from each score vector. Both of these aspects leads to a significant performance improvement over a single branch configuration. This improvement comes at the cost of a small increase in the training and prediction times. The proposed approach leads to the best performance for a given parameter budget as can be seen in tables 3 and 4, and in figure 2. Additionally, the individual "element block" performance is better as compared to when they are trained independently.

The increase in training and prediction times is mostly due to the sequential processing of branches during the forward and backward passes. The smaller size of the branches makes the data parallelism on GPUs less efficient. This effect is not as pronounced for larger models. This could be solved in two ways. First, as there is no data dependency between the branches (before the averaging layer) it is possible to extend the data parallelism to the branches, restoring the initial level of parallelism. This can be done by through a parallel implementation of multiple 2D convolutions at the same time. Second or alternatively, when multiple GPUs are used, it is possible to spread the branches over the GPUs.

Preliminary experiments on ImageNet (Russakovsky et al., 2015) show that coupled ensembles have a lower error for the same parameter budget as compared to single branch models. We will expand on these experiments in the future.

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

# Supplementary Material

## A  IMPLEMENTATION

Figure 3 shows the common structure of the test (top) and train (bottom) versions of networks used as element blocks. Figure 4 shows how it is possible in the test version to place the averaging layer just after the last FC layer of the element block instances and before the SM layer which is then "factorized". The $e$ model instances do not need to share the same architecture. Figure 5 shows how it is possible in the train version to place the averaging layer after the last FC layer, after the SM (actually LSM (LogSoftMax), which is equivalent to do a geometric mean of the SM values) layer, or after the LL layer.

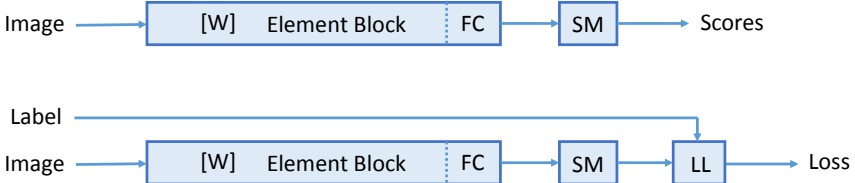

Figure 3: Versions of the element network. Top: test, bottom: train.

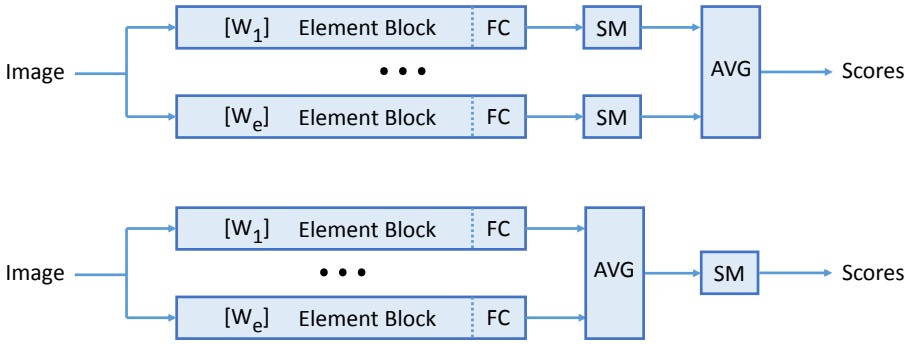

Figure 4: Test versions of coupled ensemble networks. Top: SM (classical) fusion, bottom: FC fusion. AVG: averaging layer.

We reuse "element blocks" from other groups (with appropriate credits, please let us know if any is missing or requires updating) in their original form as much as possible both for efficiency and for ensuring more meaningful comparisons. When available, we use the PyTorch implementations.

Each of the $e$ branches is defined by a parameter vector $W_e$ containing the same parameters as the original implementation. The global network is defined by a parameter vector $W$ which is a concatenation of all the $W_e$ parameter vectors. When training is done in the coupled mode and the prediction is done in a separate mode or vice-versa, a dedicated script is used for splitting the $W$ vector into the $W_e$ ones or vice-versa. In all coupled networks, for all train versions and for all test version, the same global parameter vector $W$ is used with the same split and defining the same element block functions. This is how we can combine in any way all of the four possible training conditions with all the three possible prediction conditions, even though not all of them are consistent or equally efficient.

The overall network architecture is determined by:

- the global hyper-parameter specifying the train versus test mode;
- the global hyper-parameter $e$ specifying the number of branches;
- the global hyper-parameter specifying after which layer the AVG layer should be placed (FC, SM or LL);

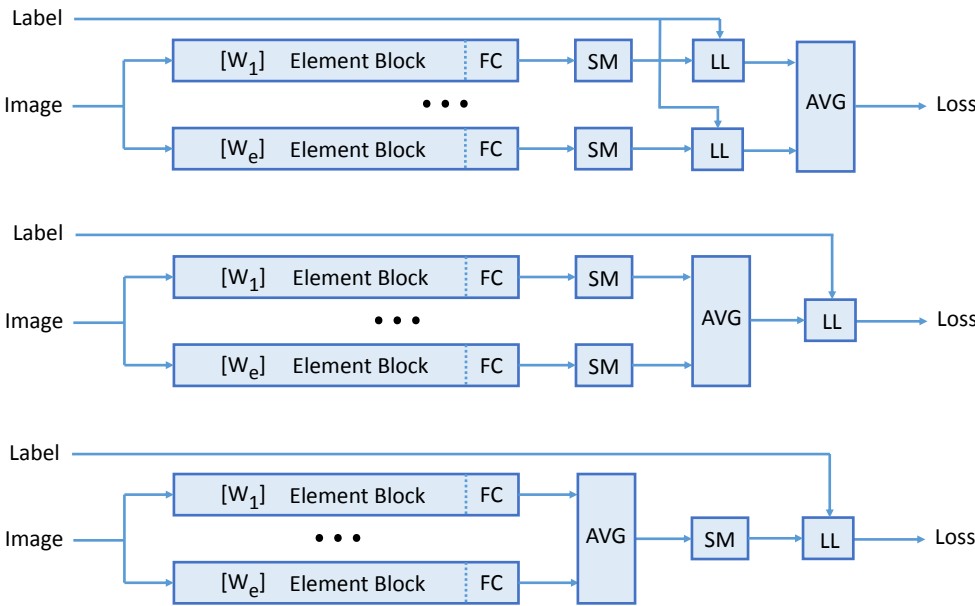

Figure 5: Train versions of coupled ensemble networks. Top: LL fusion, middle: SM fusion, bottom: FC fusion.

- either one element block to be replicated $e$ times with its own hyper-parameters or a list of $e$ element blocks, each with its own hyper-parameters.

## B  MICRO-BATCH VERSUS MINI-BATCH

For some of the larger models, it was not possible to train them with a (mini-)batch size of 64. In this case, we split data batches into $m$ "micro-batches" with $b/m$ elements each, $b$ being equal to batch size. We accumulate the gradient over these micro batches and take the average over the $m$ micro-batches to get an almost equivalent gradient, as we would have got if we processed data directly as a single batch.

The gradient would have been exactly equivalent but for the BatchNorm layer. This is because BatchNorm uses the batch mean and standard deviation to normalize the activations during the forward pass. This means that the micro-batch statistics are used whereas, ideally to get an exact equivalent, we would need the whole batch statistics. However, in practice this does not make a significant difference. Hence, to have the same settings for comparison among different models, we perform parameter updates using gradient for a batch, while performing forward passes with micro-batches (to have an optimal throughput).

In the single branch case for a given parameter budget, the need for memory depends mostly on the network depth and on the mini-batch (or micro-batch) size. We use the micro-batch "trick" for adjusting the memory need to what is available while keeping the mini-batch size equal to the default value (64 for CIFAR and SVHN). Though this is not strictly equivalent, this does not hurt performance and this would even yield an slight increase.

The multi-branch version does not require more memory if the branches' width is kept constant. More memory is needed only of the branches' width is reduced. Indeed, if we use branches with a constant parameter budget, we have to reduce either the width or the depth or both. We did some hyper-parameter search experiments by cross-validation and these indicated that the best option was a reduction of both while the exact trade-off was not very critical. In practice, for our "full-size" experiments ( 25M parameters) we did the training within the 11GB memory of GTX 1080 Ti using micro-batch sizes of 16 for the single-branch versions and of 8 for multi-branch ones. Splitting the network over two GPU boards allows for doubling the micro-batch sizes. However, this usually

does not significantly increases the speed and this does not either leads to a performance (Top-1 error rate) improvement.

## C  TEST TIME EQUIVALENCE BETWEEN FC AVERAGE AND LOGSOFTMAX AVERAGE

Given branches $E = E_1, E_2, ..E_e$, each $E_i$ produces a score vector of dimension $C$, where $C$ is the number of categories. An element of $E_i$ is referenced as $E_i^c$, were $c \in [1, C]$. FC_Average denotes averaging the raw activations from each branch. LSM_Average denotes averaging across branches, after a LogSoftMax operation in applied on each branch activation vector, separately.

Case 1: FC_average: $Scores_{FC}^c = \sum_{i=1}^{e} E_i^c$

Case 2:

$$LogSoftMax(E_n^c) = \log \frac{\exp(E_e^c)}{\sum_c \exp(E_e^c)}$$
$$= \log \exp(E_e^c) - \log \sum_c \exp(E_e^c)$$
$$= E_e^c - Z_e$$

LSM_average: $Scores_{LSM}^c = \sum_{i=1}^{e} E_i^c - \sum_{i=1}^{e} Z_i$, where $Z_e = \log \sum_C \exp(E_e^c)$. Hence we see that the LSM_average score vector is a translated version of the FC_average score vector. Also, doing an arithmetic average of LogSoftMax values is equivalent to doing a geometric average of SoftMax values. This holds during inference where we are interested only in the maximum value.

## D  ADDITIONAL RESULTS

### D.1  CHOICE OF FUSION LAYER

Table 5 shows the same results as in table 1 but for coupled ensembles with two branches only. The observations are the same as with four branches (see section 4.4). Even using only two branches provide a significant gain over a single branch architecture of comparable size.

Table 5: Coupled Ensembles of two DenseNet-BCs ($e = 2$) versus a single model of comparable complexity and study of training / prediction fusion combinations.

| L | k | e | Avg. | Individual | FC | SM | Params | Epoch(s) | Test(ms) |
|---|---|---|------|-----------|-----|-----|--------|----------|----------|
| 100 | 12 | 2 | FC | 52.68±22.95 | 22.25 | 28.78 | 1.60M | 174 | 0.98 |
| 100 | 12 | 2 | SM | 22.17±0.32 | **19.06** | 19.43 | 1.60M | 174 | 0.98 |
| 100 | 12 | 2 | LL | 22.78±0.08 | 19.33 | 19.91 | 1.60M | 174 | 0.98 |
| 100 | 12 | 2 | none | 23.13±0.15(*) | 20.44 | 20.44 | 1.60M | 171 | 0.98 |
| 100 | 17 | 1 | n/a | 21.22±0.12 | n/a | n/a | 1.57M | 121 | 0.67 |
| 124 | 14 | 1 | n/a | 21.75±0.10 | n/a | n/a | 1.55M | 135 | 0.77 |
| 148 | 12 | 1 | n/a | 20.80±0.06 | n/a | n/a | 1.56M | 159 | 0.90 |

### D.2  NUMBER OF BRANCHES

Table 6 is an extended version of table 2 in which variation of the depth $L$ and the growth rate $k$ are also evaluated for an approximately fixed parameter count. The performance is quite stable against variation of the $(L, k)$ compromise.

The same experiment was done on a validation set with a 40k/10k random split of the CIFAR-100 training set and we could draw the same conclusions from there; they led to predict that the $(L = 82, k = 8, e = 3)$ combination should be the best one on the test set. The $(L = 70, k = 9, e = 3)$ combination appeared to be slightly better here but the difference is probably not statistically significant.

Table 6: Different number of branches $e$ while varying also the depth $L$ and the growth rate $k$ for an approximately fixed parameter count.

| $L$ | $k$ | $e$ | Individual | FC | SM | Params | Epoch(s) | Test(ms) |
|---|---|---|---|---|---|---|---|---|
| 100 | 12 | 1 | **22.87±0.17(*)** | n/a | n/a | 800k | 86 | 0.51 |
| 76 | 10 | 2 | 25.58±0.20 | 21.66 | 22.17 | 720k | 103 | 0.63 |
| 88 | 9 | 2 | 25.15±0.31 | 21.87 | 22.19 | 747k | 119 | 0.71 |
| 94 | 8 | 2 | 25.72±0.20 | 21.95 | 22.22 | 666k | 115 | 0.69 |
| 100 | 8 | 2 | 25.42±0.20 | 21.87 | 22.07 | 737k | 126 | 0.75 |
| 70 | 9 | 3 | 26.67±0.40 | **21.10** | 21.24 | 773k | 129 | 0.77 |
| 82 | 8 | 3 | 26.47±0.17 | **21.25** | 21.46 | 800k | 141 | 0.85 |
| 88 | 7 | 3 | 26.92±0.41 | 22.09 | 22.49 | 698k | 148 | 0.92 |
| 94 | 7 | 3 | 26.50±0.12 | 21.95 | 22.35 | 775k | 160 | 0.98 |
| 64 | 8 | 4 | 28.58±0.59 | 22.44 | 22.58 | 719k | 142 | 0.88 |
| 70 | 8 | 4 | 27.65±0.48 | 21.50 | 22.12 | 828k | 156 | 0.94 |
| 58 | 7 | 6 | 30.11±0.53 | 23.87 | 24.22 | 718k | 179 | 1.08 |
| 64 | 7 | 6 | 30.65±0.62 | 23.08 | 23.36 | 840k | 198 | 1.20 |
| 58 | 6 | 8 | 32.15±0.00 | 25.95 | 25.70 | 722k | 219 | 1.35 |
| 64 | 6 | 8 | 31.52±0.38 | 24.42 | 24.69 | 843k | 250 | 1.51 |

## E    COMPARISON WITH *Learnt* ARCHITECTURES

In table 7, we compare the parameter usage and performance of the branched coupled ensembles with model architectures that were recovered using meta learning techniques.

Table 7: Classification error comparison with *learnt* architectures.

| System | C10+ | C100+ | SVHN | #Params |
|---|---|---|---|---|
| Neural Architecture Search v3 (Zoph & Le, 2017) | 3.65 | - | - | 37.4M |
| NASNet-A (Zoph et al., 2017) | 3.41 | - | - | 3.3M |
| DenseNet-BC $L = 82\ k = 10\ e = 4$ | 3.78 | 19.92 | 1.62 | 1.6M |
| DenseNet-BC $L = 88\ k = 14\ e = 4$ | 3.57 | 17.68 | 1.55 | 3.5M |
| DenseNet-BC $L = 88\ k = 20\ e = 4$ | 3.18 | 16.79 | 1.57 | 7.0M |

## F    PERFORMANCE MEASUREMENT AND REPRODUCIBILITY ISSUES

When attempting to compare the relative performance of different methods, we face the issue of the reproducibility of the experiments and of the statistical significance of the observed difference between performance measures. Even for a same experiment, we identified the five following sources of variation in the performance measure:

• Underlying framework for the implementation: we made experiments with Torch7 (lua) and with PyTorch.

• Random seed for the network initialization.

• CuDNN non-determinism during training: GPU associative operations are by default fast but non-deterministic. We observed that the results varies even for a same tool and the same seed. In practice, the observed variation is as important as when changing the seed.

• Fluctuations associated to the computed moving average and standard deviation in batch normalization: these fluctuations can be observed even when training with the learning rate, the SGD momentum and the weight decay all set to $0$. During the last few epochs of training, their level of influence is the same as with the default value of these hyper-parameters.

- Choice of the model instance chosen from training epochs: the model obtained after the last epoch, or the best performing model. Note that choosing the best performing model involves looking at test data.

Regardless of the implementation, the numerical determinism, the Batch Norm moving average, and the epoch sampling questions, we should still expect a dispersion of the evaluation measure according to the choice of the random initialization since different random seeds will likely lead to different local minima. It is generally considered that the local minima obtained with "properly designed and trained" neural networks should all have similar performance (Kawaguchi, 2016). We do observe a relatively small dispersion (quantified by the standard deviation below) confirming this hypothesis. This dispersion may be small but it is not negligible and it complicates the comparisons between methods since differences in measures lower than their dispersions is likely to be non-significant. Classical statistical significance tests do not help much here since differences that are statistically significant in this sense can be observed between models obtained just with different seeds (and even with the same seed), everything else being kept equal.

Experiments reported in this section gives an estimation of the dispersion in the particular case of a moderate scale model. We generally cannot afford doing a large number of trials for larger models.

We tried to quantify the relative importance of the different effects in the particular case of DenseNet-BC with $L = 100, k = 12$ on CIFAR 100. The upper part of table 8 shows the results obtained for the same experiment in the four groups of three rows. We tried four combinations corresponding to the use of Torch7 versus PyTorch and to the use of the same seed versus the use of different seeds. For each of these configuration, we used as the performance measure: (i) the error rate of the model computed at the last epoch or (ii) the average of the error rate of the models computed at the last 10 epochs, (iii) the error rate of the model having the lowest error rate over all epochs. For these $2 \times 2 \times 3$ cases, we present the minimum, the median, the maximum and the mean±standard deviation over 10 measures corresponding to 10 identical runs (except for the seed when indicated). Additionally, in the case of the average of the error rate of the models computed at the 10 last epochs, we present the root mean square of the standard deviation of the fluctuations on the last 10 epochs (which is the same as the square root of the mean of their variance). We make the following observations:

- There does not seem to be a significant difference between Torch7 and PyTorch implementations;
- There does not seem to be a significant difference between using a same seed and using different seeds; the dispersion observed using the same seed (with everything else being equal) implies that there is no way to exactly reproduce results;
- There does not seem to be a significant difference between the means over the 10 measures computed on the single last epoch and the means over the 10 measures computed on the last 10 epochs;
- The standard deviation of the measures computed on the 10 runs is slightly but consistently smaller when the measures are computed on the last 10 epochs than when they are computed on the single last epoch; this is the same for the difference between the best and the worst measures; this was expected since averaging the measure on the last 10 epochs reduces the fluctuations due to the moving average and standard deviation computed in batch normalization and possibly too the the random fluctuations due to the final learning steps;
- The mean of the measures computed on the 10 runs is significantly lower when the measure is taken at the best epoch than when they are computed either on the single last epoch or on the last 10 epochs. This is expected since the minimum is always below the average. However, presenting this measure involves using the test data for selecting the best model.

Following these observations, we propose a method for ensuring the best reproducibility and the fairest comparisons. Choosing the measure as the minimum of the error rate for all models computed during the training seems neither realistic nor a good practice since we have no way to know which model will be the best one without looking at the results (cross-validation cannot be used for that) and this is like tuning on the test set. Even though this is not necessarily unfair for system comparison if the measures are done in this condition for all systems, this does introduce a bias for the absolute performance estimation. Using the error rate at the last iteration or at the 10 last iteration does not seem to make a difference in the mean but the standard deviation is smaller for the latter, therefore this one should be preferred when a single experiment is conducted. We also checked that using the 10 or the 25 last epochs does not make much difference (learning at this point does not seem to

Table 8: Performance measurement and reproducibility issues. Statistics on 10 runs.

| Seeds | Impl. | Last | $L$ | $k$ | $e$ | Min. | Med. | Max. | Mean±SD | RMS(SD) |
|-------|-------|------|-----|-----|-----|------|------|------|---------|---------|
| diff. | PyT. | 1 | 100 | 12 | 1 | 22.64 | 22.80 | 23.22 | 22.89±0.21 | n/a |
| diff. | PyT. | 10 | 100 | 12 | 1 | 22.67 | 22.83 | 23.14 | 22.87±0.17 | 0.13 |
| diff. | PyT. | best | 100 | 12 | 1 | 22.13 | 22.56 | 22.91 | 22.54±0.24 | n/a |
| same | PyT. | 1 | 100 | 12 | 1 | 22.77 | 23.05 | 23.55 | 23.06±0.23 | n/a |
| same | PyT. | 10 | 100 | 12 | 1 | 22.81 | 22.98 | 23.49 | 23.04±0.22 | 0.11 |
| same | PyT. | best | 100 | 12 | 1 | 22.44 | 22.67 | 23.02 | 22.71±0.18 | n/a |
| diff. | LuaT. | 1 | 100 | 12 | 1 | 22.55 | 22.94 | 23.11 | 22.90±0.20 | n/a |
| diff. | LuaT. | 10 | 100 | 12 | 1 | 22.55 | 22.89 | 23.08 | 22.86±0.20 | 0.12 |
| diff. | LuaT. | best | 100 | 12 | 1 | 22.17 | 22.52 | 22.75 | 22.49±0.18 | n/a |
| same | LuaT. | 1 | 100 | 12 | 1 | 22.33 | 22.82 | 23.58 | 22.82±0.34 | n/a |
| same | LuaT. | 10 | 100 | 12 | 1 | 22.47 | 22.92 | 23.51 | 22.87±0.30 | 0.12 |
| same | LuaT. | best | 100 | 12 | 1 | 22.24 | 22.51 | 23.24 | 22.54±0.29 | n/a |
| diff. | PyT. | 1 | 82 | 8 | 3 | 21.27 | 21.44 | 21.70 | 21.49±0.15 | n/a |
| diff. | PyT. | 10 | 82 | 8 | 3 | 21.24 | 21.46 | 21.63 | 21.45±0.11 | 0.12 |
| diff. | PyT. | best | 82 | 8 | 3 | 20.84 | 21.18 | 21.30 | 21.14±0.14 | n/a |
| diff. | PyT. | 1 | 100 | 12 | 4 | 17.24 | 17.71 | 17.86 | 17.65±0.18 | n/a |
| diff. | PyT. | 10 | 100 | 12 | 4 | 17.37 | 17.67 | 17.81 | 17.66±0.14 | 0.11 |
| diff. | PyT. | best | 100 | 12 | 4 | 17.11 | 17.46 | 17.66 | 17.45±0.16 | n/a |
| diff. | PyT. | 1 | 106 | 33 | 4 | 15.47 | 15.80 | 16.22 | 15.83±0.23 | n/a |
| diff. | PyT. | 10 | 106 | 33 | 4 | 15.51 | 15.84 | 16.14 | 15.85±0.22 | 0.10 |
| diff. | PyT. | best | 106 | 33 | 4 | 15.33 | 15.54 | 15.83 | 15.61±0.18 | n/a |

lead to further improvement). A value different from 10 can be used and this is not critical. In all the CIFAR experiments reported in this paper, we used the average of the error rate for the models obtained at the last 10 epochs as this should be (slightly) more robust and more conservative. The case for SVHN experiments is slightly different since there is a much smaller number of much bigger epochs; we used the last 4 iterations in this case.

These observations have been made in a quite specific case but the principle and the conclusions (use of the average of the error rate from the last epochs should lead to more robust and conservative results) are likely to be general. Table 8 also shows the results for a coupled ensemble network of comparable size, for a coupled ensemble network four times bigger, and for a coupled ensemble network approximately 32 times bigger. Similar observations can be made and, additionally, we can observe that both the range and the standard deviations are smaller for networks of comparable sizes. This might be because an averaging is already made between the branches leading to a reduction of the variance.

## G   TRAINING TIME BUDGET

In this study, all comparisons between single-branch (classical) and multi-branch architectures have been made at constant parameter budget and they have shown a clear advantage for the multi-branch networks under this condition. However, the training time of multi-branch networks is currently significantly longer than the training time of the corresponding single branch network. Though this might be improved by better parallelization, we investigate here whether the multi-branch architectures can still improve over single-branch ones at constant training time budget with the current implementation.

There are several ways to reduce the training time: (i) reduce the number of iterations; (ii) reduce the parameter count; and/or (iii) increase the width while reducing the depth for a constant parameter count. Table 9 shows the results obtained for these three options while taking as a baseline, the single branch DenseNet-BC $L = 190$, $k = 40$, $e = 1$, which has a training time of about 80 hours using a single NVIDIA GTX 1080Ti. Results are shown for CIFAR 10 and 100 with statistics on 5 runs for each configuration. The corresponding multi-branch baseline with the same parameter

budget is DenseNet-BC $L = 106$, $k = 33$, $e = 4$, whose training time for 300 epochs is about 1.6 times that of the baseline's. Option (i) correspond to reducing the number of training epochs to 188; option (ii) corresponds to reducing the depth $L$ from 106 to 88, thereby reducing the parameter count (reducing the width increases the training time); and option (iii) matches both the parameter count and training time of the single-branch baseline with DenseNet-BC $L = 76$, $k = 44$, $e = 4$. Additionally, results are shown for DenseNet-BC $L = 94$, $k = 26$, $e = 4$ and for DenseNet-BC $L = 88$, $k = 20$, $e = 4$, which respectively correspond to a parameter budget divided by 2 and by 3.7.

Table 9: Performance versus training time and network size (GTX 1080Ti), statistics on 5 runs.

| L | k | e | Params. | Epochs | Train time (h) | C10 | C100 |
|---|---|---|---------|--------|----------------|-----|------|
| 190 | 40 | 1 | 25.8M | 300 | 79.3 | 3.75±0.19 | 17.31±0.18 |
| 106 | 33 | 4 | 25.1M | 300 | 126.6 | 3.02±0.03 | 15.76±0.10 |
| 106 | 33 | 4 | 25.1M | 188 | 79.3 | 3.14±0.02 | 16.20±0.10 |
| 88 | 33 | 4 | 18.6M | 300 | 79.9 | 3.05±0.06 | 16.09±0.15 |
| 76 | 44 | 4 | 25.8M | 300 | 79.9 | 3.05±0.09 | 16.06±0.17 |
| 94 | 26 | 4 | 13.0M | 300 | 67.1 | 3.13±0.09 | 16.27±0.16 |
| 88 | 20 | 4 | 7.0M | 300 | 52.3 | 3.32±0.04 | 16.87±0.15 |

Even though they perform slightly worse than the full multi-branch baseline, all three options still perform significantly better than the single-branch baseline (with differences between them possibly not statistically significant). DenseNet-BC $L = 88$, $k = 20$, $e = 4$ still performs better than the single-branch baseline with a parameter count divided by 3.7 and a training time divided by 1.5.

## H  TRAINING DATA EFFICIENCY

In this section we compare the performance of single branch models and coupled ensembles in the context of a low training data scenario. We train a single branch model and multi-branch coupled ensemble, both having the same number of parameters, on two datasets: STL-10 (Coates et al., 2011) and a 10K balanced random subset of the 50K CIFAR-100 training set. STL-10 comprises images of size 96x96, grouped into 10 classes. Each class has 500 training images. The test set has 8000 images. Results are shown in table 10. We see that for a fixed parameter budget, coupled ensembles significantly outperform the single branch model.

Table 10: Results with low training data.

| Dataset | Single Branch | Coupled Ensemble | #Params |
|---------|---------------|------------------|---------|
| STL-10 | 41.73 | 32.01 | 0.8M |
| CIFAR-100 10K subset | 41.84 | 36.30 | 3.2M |

## I  PRELIMINARY EXPERIMENTS ON IMAGENET

We conducted preliminary experiments on ILSVRC2012 (Russakovsky et al., 2015) to compare single-branch model and multi-branch coupled ensembles. Due to time, storage and memory constraints on our machines, these experiments were done on images of size 256×256, instead of the original sized images. Data augmentation involved only random horizontal flips, and random crops of size 224×224.

For a baseline single-branch model, we use DenseNet-169-k32-e1 and we compare this with coupled ensemble DenseNet-121-k30-e2. We realize this is not a state-of-the-art baseline but due to the constraints we had this was the strongest possible. We are carrying out experiments with full sized images and increased data augmentation. We will update the manuscript when they become available but they will be after the deadline. The current results are displayed in table 11 and they show that

the coupled ensemble approach with two branches produces a significant improvement over the baseline, even wit a constant training time budget.

Table 11: Preliminary results on ImageNet.

| L | k | e | Params. | Epochs | Train time (h) | Top-1 error |
|---|---|---|---------|--------|----------------|-------------|
| 161 | 32 | 1 | 14.1M | 90 | 162 | 31.21 |
| 121 | 30 | 2 | 14.1M | 90 | 225 | 29.41 |
| 121 | 30 | 2 | 14.1M | 64 | 160 | 29.83 |

