# OpenReview forum: "Coupled Ensembles of Neural Networks"
_ICLR.cc/2018/Conference — Reject_

### Official Review · AnonReviewer2 · 2017-11-25
**This work proposed a reconfiguration of the existing state-of-the-art CNN model using a new branching architecture.**

**Rating:** 6
**Confidence:** 4

**Review:**

This work proposed a reconfiguration of the existing state-of-the-art CNN model architectures including ResNet and DensNet. By introducing new branching architecture, coupled ensembles, they demonstrate that the model can achieve better performance in classification tasks compared with the single branch counterpart with same parameter budget. Additionally, they also show that the proposed ensemble method results in better performance than other ensemble methods (For example, ensemble over independently trained models)  not only in combined mode but also in individual branches.

Paper Strengths:
* The proposed coupled ensembles method truly show impressive results in classification benchmark (DenseNet-BC L = 118 k = 35 e = 3).
* Detailed analysis on different ensemble fusion methods on both training time and testing time.
* Simple but effective design to achieve a better result in testing time with same total parameter budget.

Paper Weakness:
* Some detail about different fusing method should be mentioned in the main paper instead of in the supplementary material.
* In practice, how much more GPU memory is required to train the model with parallel branches (with same parameter budgets) because memory consumption is one of the main problems of networks with multiple branches.
* At least one experiment should be carried out on a larger dataset such as ImageNet to further demonstrate the validity of the proposed method.
* More analysis can be conducted on the training process of the model. Will it converge faster? What will be the total required training time to reach the same performance compared with single branch model with the same parameter budget?

---

> ### Author Response · Authors · 2018-01-05
> **Responses to Reviewer 2.**
>
> Thank you for the review and valuable feedback. Please find our responses to your questions below:
>
> 1. Some detail about different fusing method should be mentioned in the main paper instead of in the supplementary
> material.
>
> Details are given in the supplementary material but the fusion methods are also discussed in section 3. A figure has
> also been inserted to demonstrate the architecture (Figure 1).
>
>
> 2. In practice, how much more GPU memory is required to train the model with parallel branches (with same parameter
> budgets) because memory consumption is one of the main problems of networks with multiple branches.
>
> A discussion of the memory requirements and how we address it has been added to section B of supplementary material.
>
>
> 3. At least one experiment should be carried out on a larger dataset such as ImageNet to further demonstrate the
> validity of the proposed method.
>
> We have started experiments on ImageNet, the results are reported in Section I. Results show a benefit from using
> coupled ensembles. Currently the baseline is not state-of-the-art. We are conducting additional experiments and will
> update when they are available.
>
>
> 4. More analysis can be conducted on the training process of the model. Will it  converge faster? What will be the
> total required training time to reach the same performance compared with single branch model with the same parameter
> budget?
>
> The multi-branch approach leads to better performance even with a constant training time budget. We have added section G in the supplementary material with new experimental results and a discussion.

---

### Official Review · AnonReviewer1 · 2017-11-27
**simple approach, shows parameter-saving benefits of coupled ensembling**

**Rating:** 6
**Confidence:** 4

**Review:**

Strengths:
* Very simple approach, amounting to coupled training of "e" identical copies  of a chosen net architecture, whose predictions are fused during training. This forces the different model instances to become more complementary.
* Perhaps counterintuitively, experiments also show that coupled ensembling leads to individual nets that perform better than those produced by separate training.
* The practical advantages of the proposed approach are twofold:
1. Given a fixed parameter budget, coupled ensembling leads to better accuracy than a single net or an ensemble of disjointly-trained nets.
2. For the same accuracy, coupled ensembling yields significant parameter savings.

Weaknesses:
* Although results are very strong, the proposed models do not outperform the state-of-the-art, except for the models reported in Table 4, which however were obtained by *traditional* ensembling of coupled ensembles.
* Coupled ensembling requires joint training of all nets in the ensemble and thus is limited by the size of the model that can be fit in memory. Conversely, traditional ensembling involves separate training of the different instances and this enables the learning of an arbitrary number of individual nets.
* I am surprised by the results in Table 2, which suggest that the optimal number of nets in the ensemble is remarkably low (only 3!). It'd be valuable to understand whether this kind of result holds for other network architectures or whether it is specific to this choice of net.
* Strictly speaking it is correct to refer to the individual nets in the ensembles as "branches" and "basic blocks." Nevertheless, I find the use of these terms confusing in the context of the proposed approach, since they are commonly used to denote concepts different from those represented here.  I would recommend refraining from using these terms here.

Overall, the paper provides limited technical novelty. Yet, it reveals some interesting empirical findings about the benefits of coordinated training of models in an ensemble.

---

> ### Author Response · Authors · 2018-01-05
> **Responses to Reviewer 1.**
>
> Thank you for the review and valuable feedback. Please find our responses to your questions below:
>
> 1. Although results are very strong, the proposed models do not outperform the state-of-the-art, except for the
> models reported in Table 4, which however were obtained by *traditional* ensembling of coupled ensembles.
>
> We found two works which achieve better performances:
>
> Cutout regularization: This is a data augmentation scheme which is applied to existing models and improves their
> performance. It is likely that cutout applied to coupled ensembles will also lead to better performance. In contrast, the proposed coupled ensembles scheme applies to the model architecture itself.
>
> ShakeDrop: Modification of previously proposed Shake-Shake method. We propose an architectural deisgn choice. Similar to 'cutout', it is likely that ShakeDrop can be adapted to the coupled ensemble framework, leading to improved performance.
>
> Apart from these two works and as far as we know, our results are on par with or better than the state of the art for all parameter budget.
>
>
> 2. Coupled ensembling requires joint training of all nets in the ensemble and thus is limited by the size of the
> model that can be fit in memory. Conversely, traditional ensembling involves separate training of the different
> instances and this enables the learning of an arbitrary number of individual nets.
>
> Coupled ensemble learning is precisely a way to increase the performance (minimizing the top-1 error rate) for a
> given parameter budget (and/or for a given memory budget, see Section B), and/or for a given training time budget (
> see section G). As we report in section 4.7, it is possible to use the classical ensemble learning approach on top of
> the coupled ensemble learning one to obtain further benefit.
>
>
> 3. I am surprised by the results in Table 2, which suggest that the optimal number of nets in the ensemble is
> remarkably low (only 3!). It'd be valuable to understand whether this kind of result holds for other network
> architectures or whether it is specific to this choice of net.
>
> The optimum number probably depends on the the network architecture, on the target task, and on the network size. The
> network size is likely to have a strong influence. The target network size in table 2 is of only 0.8M. On the other
> hand, from table 3, we can see that when the target network size is 32 times bigger, the difference in overall
> performance is not statistically significant (see section F for number of branches varying from 3 to 8.
>
>
> 4. Strictly speaking it is correct to refer to the individual nets in the ensembles as "branches" and "basic blocks."
> Nevertheless, I find the use of these terms confusing in the context of the proposed approach, since they are
> commonly used to denote concepts different from those represented here.  I would recommend refraining from using
> these terms here.
>
> Yes, this is a problem for which we have not yet found a good solution. "Instance", "element" or "column" could be
> used too. We changed "basic" to "element" as this is consistent with the ensembling terminology but we kept "branch"
> as it actually correspond to the high-level network architecture. We understand that this might be confusing since
> the internal structure of the element blocks may already be branched (e.g. ResNeXt or Shake-Shake) but this risk of
> confusion is limited in practice at the level of granularity that we are considering here.

---

### Official Review · AnonReviewer3 · 2017-12-06
**Branched architecture with early split and late fusion have benefits over a single branch architecture with same number of parameters.**

**Rating:** 6
**Confidence:** 4

**Review:**

This paper presents a deep network architecture which processes data using multiple parallel branches and combines the posterior from these branches to compute the final scores; the network is trained in end-to-end, thus training the parallel branches jointly. Existing literature with branching architecture either employ a 2 stage training approach, training branches independently and then training the fusion network, or the branching is restricted to local regions (set of contiguous layers). In effect, this paper extends the existing literature suggesting end-to-end branching. While the technical novelty, as described in the paper, is relatively limited, the thorough experimentation together with detailed comparisons between intuitive ways to combine the output of the parallel branches is certainly valuable to the research community.

+ Paper is well written and easy to follow.
+ Proposed branching architecture clearly outperforms the baseline network (same number of parameters with a single branch) and thus offer yet another interesting choice while creating the network architecture for a problem
+ Detailed experiments to study and analyze the effect of various parameters including the number of branches as well as various architectures to combine the output of the parallel branches.
+ [Ease of implementation] Suggested architecture can be easily implemented using existing deep learning frameworks.

- Although joint end-to-end training of branches certainly brings value compared to independent training, but the increased resource requirements may limits the applicability to large benchmarks such as ImageNet. While authors suggests a way to circumvent such limitations by training branches on separate GPUs but this would still impose limits on the number of branches as well as its ease of implementation.
- Adding an overview figure of the architecture in the main paper (instead of supplementary) would be helpful.
- Branched architecture serve as a regularization by distributing the gradients across different branches; however this also suggests that early layers on the network across branches would be independent. It would helpful if authors would consider an alternate archiecture where early layers may be shared across branches, suggesting a delayed branching, with fusion at the final layer.
- One of the benefits of architectures such as DenseNet is their usefulness as a feature extractor (output of lower layers) which generalizes even to domain other that the dataset; the branched architecture could potentially diminish this benefit.

Minor edits: Page 1. 'significantly match and improve' => 'either match or improve'

Additional notes:
- It would interesting to compare this approach with a conditional training pipeline that sequentially adds branches, keeping the previous branches fixed. This may offer as a trade-off between benefits of joint training of branches vs being able to train deep models with several branches.

---

> ### Author Response · Authors · 2018-01-05
> **Responses to Reviewer 3.**
>
> Thank you for the review and valuable feedback. Please find our responses to your questions below:
>
> 1. Although joint end-to-end training of branches certainly brings value compared to independent training, but the
> increased resource requirements may limits the applicability to large benchmarks such as ImageNet. While authors
> suggests a way to circumvent such limitations by training branches on separate GPUs but this would still impose
> limits on the number of branches as well as its ease of implementation.
>
> Coupled ensemble learning is precisely a way to increase the performance (minimizing the top-1 error rate) for a
> given parameter budget (and/or for a given memory budget, see section B) and/or for a given
> training time budget (see section G)). Regarding the training on multiple GPUs, branch parallelism
> is a quite natural and efficient way to split the storage and to parallelize the computation but this is not the only
> possible one. Also, our experiments suggest that the network performance does not critically depends on the exact
> number of branches.
>
>
> 2. Adding an overview figure of the architecture in the main paper (instead of supplementary) would be helpful.
>
> A figure has been inserted in section 3.
>
>
> 3. Branched architecture serve as a regularization by distributing the gradients across different branches; however
> this also suggests that early layers on the network across branches would be independent. It would helpful if authors
> would consider an alternate architecture where early layers may be shared across branches, suggesting a delayed
> branching, with fusion at the final layer.
>
> Thanks for the suggestion. We planned to investigate this but we did not have enough time before the deadline and
> the paper is already quite long.
>
>
> 4. One of the benefits of architectures such as DenseNet is their usefulness as a feature extractor (output of lower
> layers) which generalizes even to domain other that the dataset; the branched architecture could potentially diminish
> this benefit.
>
> We see no a priori reason why features extracted by branched architecture should be less efficient than those
> extracted from non-branched ones. We even see no a priori reason either why the benefit they bring in classification
> tasks should not be transferred also with the extracted features. We will conduct such transfer
> experiments in the future.
>
>
> 5. It would interesting to compare this approach with a conditional training pipeline that sequentially adds
> branches, keeping the previous branches fixed. This may offer as a trade-off between benefits of joint training of
> branches vs being able to train deep models with several branches
>
> Thanks for the suggestion. We had planned to investigate this but we did not have enough time before the deadline.

---

### Author Response · Authors · 2018-01-05
**Changer made to the paper during the rebuttal phase.**

We have updated the paper based on the reviewer suggestions and also added responses to their questions.

Main updates:

- added figure to demonstarte the model architecture and fusion scheme (Figure 1)
- added Section G to compare between single-branch and multi-branch models for a fixed training time budget.
- added Section H to compare between single-branch and multi-branch models in a low training data scenario.
- added Section I for experiments on ImageNet

Update in table 9 on January 11.

---

### Decision · Program_Chairs · 2018-01-29
**ICLR 2018 Conference Acceptance Decision**

**Decision:**

Reject

**Comment:**

The paper studies end-to-end training of a multi-branch convolutional network. This appears to lead to strong accuracies on the CIFAR and SVHN datasets, but it remains unclear whether or not this results transfers to ImageNet. The proposed approach is hardly novel, and lacks a systematic comparison with "regular" ensembling methods and with related mixture-of-experts approaches (for instance: S. Gross et al. Hard Mixtures of Experts for Large Scale Weakly Supervised Vision, 2017; Shazeer et al. Outrageously Large Neural Networks: The Sparsely-Gated Mixture-of-Experts Layer, 2017).